# Geometrical Parameter Effect on Plasmonic Scattering of Bimetallic Three-Layered Nanoshells

**DOI:** 10.3390/nano12213816

**Published:** 2022-10-28

**Authors:** Ji-Bao Hu, Yu-Lin Chen, Juan Li, Ye-Wan Ma, Chuan-Cun Shu

**Affiliations:** 1School of Physics and Electric Engineering, Anqing Normal University, Anqing 246011, China; 2School of Physics and Optoelectronic Engineering, Nanjing University of Information Science and Technology, Nanjing 210044, China; 3Hunan Key Laboratory of Nanophotonics and Devices, Hunan Key Laboratory of Super-Microstructure and Ultrafast Process, School of Physics and Electronics, Central South University, Changsha 410083, China

**Keywords:** bimetallic three-layered nanoshells, scattering cross section, scattering efficiency, quasi-static theory

## Abstract

Enhanced scattering from local surface plasmon resonance by light has attracted much attention due to its special applications in sensor, cell, and biological imaging. Here, we investigate the ratio of scattering to absorption in bimetallic three-layered nanoshells with different geometrical parameters using quasi-static theory. We show that the ratio of scattering to absorption strongly depends on the inner radius, shell thickness, middle dielectric function, and surrounding medium function. To gain insight into the effect of such geometrical parameters on the plasmonic scattering, we also provide a comparison between silver–dielectric–gold nanoshells and gold–dielectric–silver nanoshells. This work provides an alternative approach to analyze the optical properties of bimetallic three-layered nanoshells with potential applications in sensors and photo-detectors.

## 1. Introduction

Due to their unique optical and optoelectrical properties, the local surface plasmon resonance (LSPR) of noble metal nanostructure induced by the collective motion of conduction electrons has attracted widespread attention from surface-enhanced Raman scattering (SERS) [1,2], sensor [3,4], cell, and biological imaging [5,6]. Different metallic nanostructures are designed theoretically and experimentally to achieve wavelength matching for specific applications [7,8,9,10,11]. LSPR can be well tuned from the ultraviolet to the infrared region by adjusting its geometrical parameter and surrounding medium function [12,13]. Unlike solid nanoparticles, noble nanoshells exhibit tunable plasmon resonance with minimal absorption in the near-infrared region by adjusting its geometrical parameters. Plasmon hybridization theory can explain the physical mechanism of nanoshells well [14,15], and the interaction between the plasmon responses of a solid sphere and a cavity will produce two new plasmon modes (the higher energy and the lower energy). Xia et al. have synthesized concentric nanoshells with a diameter of 50nm, containing a gold core, a tunable silicon dielectric layer, and an outer gold nanoshell [16]. Theoretically, Khosravi et al. have studied the light scattering from two alternating concentric double silica-gold nanoshells by using quasi-static theory. The result has demonstrated that the intensity and position of the scattering peaks depend on shell thicknesses [17]. Shirzaditabar et al. have studied geometrical parameters’ effect on the local electric field of silver–dielectric–silver multi-layer nanoshell; it is found that the resonance wavelength exhibits a red-shift by increasing the silver core radius [18,19]. Campbell and Ziolkowski have designed concentric double-negative metamaterial spheres. The results show that absorption can be overcome, and new optical properties can be observed in coated shell particles. In addition, the coated nanoparticle can be applied in the design of highly subwavelength optical amplifiers [20,21,22,23]. Zouros and Tsakmakidis have presented a high permittivity semiconductor structure capable of breaking the scattering efficiency’s fundamental single-channel limit and demonstrated the feasibility of obtaining magnetically-tunable directionality inversion in spherical microresonators (THz antennas) coated by magneto-optical materials [24,25].

Although both the scattering and absorption result from LSPR, the intensity of light scattering is usually less than that of light absorption. Thus, improving the ratio of scattering to absorption is an essential research topic. To extract the contribution of scattering from LSPR, Daneshfar et al. have studied the effect of various plasmonic materials on the scattering-to-absorption ratio of metal-dielectric two-layered nanoshell with quasi-static theory. The result shows that the ratio of scattering to absorption strongly depends on the shape and different materials [26]. Using Mie’s theory, Bansal et al. have examined the absorption and scattering of semiconductor-coated nanoshells. They found that absorption and scattering were blue-shifting with increasing core radius, and red-shattering with increasing surrounding medium function [27]. A recent study on the scattering of $Ag_{1-x}-Cu_{x}$, $Ag_{1-x}-Au_{x}$ and $Au_{1-x}-Cu_{x}$ alloy nanoshell has demonstrated that the scattering enhancement can be controlled and tuned by adjusting the composition and surrounding medium [28]. Using quasi-static theory, Zhu et el. have found that the maximum effective scattering occurs at a thick thickness of nanoshell due to the plasmon mode transformation [29].

In this work, we theoretically examine the geometrical parameters’ effect on plasmonic scattering of bimetallic three-layered nanoshells. We show that the ratio of absorption to scattering depends on its geometrical parameters and surrounding medium function. Furthermore, our simulations demonstrate that the ratio of scattering to absorption can be improved by increasing the inner core radius to a suitable wavelength and will be reduced by increasing the thickness of the middle dielectric layer and middle dielectric function. This work has the potential applications of bimetallic three-layered nanoshells in SERS, biological tissues, cells, and sensors.

## 2. Calculation Model and Theoretical Method

The geometry of bimetallic three-layered nanoshells is shown in Figure 1, in which the inner nanosphere has a a radius $r_{1}$ and a complex dielectric function $\varepsilon_{1}$, the outer nanoshell has a thickness of $r_{3}-r_{2}$ and a complex dielectric function $\varepsilon_{3}$, the inner nanosphere and outer nanoshell are separated by a dielectric layer with a thickness of $r_{2}-r_{1}$ and a dielectric function $\varepsilon_{2}$, and the total geometry is suspended in surrounding medium function $\varepsilon_{4}$. The total diameter of the nanoshell is much smaller than the incident wavelength. Thus, the quasi-static theory can be used to calculate the scattering and absorption efficiency [29,30], and the electric potential that satisfies Laplace’s equation reads
(1)$$\nabla^{2}\varphi =0.$$

In the spherical coordinate system, the solution of Equation (Equation 1) with the electric potential $\varphi_{i}\left(r,\theta \right)$ in each region can be given by [31]
(2)$$\varphi_{i}\left(r,\theta \right)=\left(A_{i}r+B_{i}r^{-2}\right)\cos\theta ,\left(i=1,\ldots ,4\right)$$
where *r* and θ correspond to the radial distance and the polar angle, respectively, and the coefficients $A_{i}$ and $B_{i}$ are determined by satisfying the following electromagnetic field boundary conditions. That is, the electric potential should be continued at the adjacent surface
(3)$$\varphi_{i}\left(r,\theta \right){{|}_{r=r_{i}}=\varphi_{i+1}\left(r,\theta \right)|}_{r=r_{i}},$$
and the normal derivative of electric potential at the surface should satisfy
(4)$$\varepsilon_{i}\frac{\partial \varphi_{i}\left(r,\theta \right)}{\partial r}{{|}_{r=r_{i}}=\varepsilon_{i+1}\frac{\partial \varphi_{i+1}\left(r,\theta \right)}{\partial r}|}_{r=r_{i}}$$
because of the continuity of the normal component of the displacement field at the boundary.

Considering the natural boundary conditions, in the inner core region i=1, $B_{1}=0$. In the surrounding medium region i=4, far from the nanoshell, the electric potential can be expressed as $\varphi_{4}=-E_{0}r\cos\theta$, where the magnitude of the incident electric field $E_{0}=-A_{4}$. The other coefficients $A_{1}$, $A_{2}$, $A_{3}$ and $B_{2}$, $B_{3}$, $B_{4}$ can be derived from Equations (Equation 3) and (Equation 4). By using the gradient of the electric potential, the electric field intensity E in each region can be obtained
(5)$$E_{i}\left(r,\theta \right)=-\nabla \varphi_{i}\left(r,\theta \right)=-\left(A_{i}-2B_{i}r^{-3}\right)\cos\theta \vec{e_{r}}+\left(A_{i}+B_{i}r^{-3}\right)\sin\theta \vec{e_{\theta }},\left(i=1,\ldots ,4\right).$$

Since the particle is much smaller than wavelength, the induced field in the region outside the shell is the same as a dipole with an effective dipole moment P ($P=\varepsilon_{4}\alpha E_{4}$), where α is the polarizability. As a result, the scattering and absorption cross section can be derived by using optical scattering theory [32]
(6)$$C_{sca}=\frac{k^{4}{\left|\alpha \right|}^{2}}{6\pi {\varepsilon_{0}}^{2}}$$
(7)Cabs=kIm(α)
where *k* is the wave number, $\varepsilon_{0}$ is the permittivity of free space, and Im is imaginary part. In this paper, the ratio of scattering to absorption (scattering efficiency) is defined as $\frac{C_{sca}}{C_{abs}}$.

The complex dielectric functions of gold and silver have real and imaginary frequency-dependent components. In the Drude model [33], the complex dielectric function reads
(8)$$\varepsilon \left(\omega \right)=\varepsilon_{\infty }-\frac{{\omega_{p}}^{2}}{\omega^{2}+i\omega \gamma }$$
where $\varepsilon_{\infty }$ is the value of the dielectric constant at high frequency, $\omega_{p}$ is the bulk plasmon frequency, ω is the angular frequency of the incident field, and γ is the damping constant from the scattering of noble metal electrons. In this calculation, the values of parameters for silver are $\gamma_{Ag}=3.2258\times 10^{13}s^{-1}$, $\varepsilon_{\infty }^{Ag}=4.039$, and $\omega_{p}^{Ag}=9.1721\mathrm{eV}$, and for gold $\gamma_{Au}=1.07\times 10^{14}s^{-1}$, $\varepsilon_{\infty }^{Au}=8.7499$, and $\omega_{p}^{Au}=9.0146\mathrm{eV}$.

## 3. Numerical Results and Discussion

Figure 2 shows the spectra of light absorption, scattering, and scattering efficiency as a function of wavelength, where the geometric parameters are $\left[r_{1},r_{2},r_{3}\right]$ = [10 nm, 25 nm, 30 nm] and $\left[\varepsilon_{2},\varepsilon_{4}\right]=\left[2.5,1.77\right]$. By examining the spectra of silver–dielectric–silver three-layered nanoshell as a function of wavelength in Figure 2a, we can see that three peaks $|\omega_{-}^{-}\rangle$, $|\omega_{-}^{+}\rangle$ and $|\omega_{+}^{-}\rangle$ appear in the spectra, caused by the plasmon hybridization [14,15,17,18]. Experimentally, the two peaks at longer wavelengths $|\omega_{-}^{-}\rangle$ and $|\omega_{-}^{+}\rangle$ can be easily detected. We can also see two longer absorption wavelength peaks at λ=648 nm and λ=412 nm. The longest wavelength λ=648 nm, denoted as $|\omega_{-}^{-}\rangle$, corresponds to an anti-symmetric coupling between the solid inner sphere and outer bonding nanoshell. The peak at the wavelength λ=412 nm, denoted as $|\omega_{-}^{+}\rangle$, is caused by a symmetric coupling between the solid inner sphere and outer bonding nanoshell. The corresponding scattering peaks are located at λ=650 nm ($|\omega_{-}^{-}\rangle$) and λ=412 nm ($|\omega_{-}^{+}\rangle$). Unlike the intensity of absorption, the intensity of scattering intensity is much weaker than that of absorption. Therefore, the scattering efficiency as a function of wavelength in Figure 2a corresponds to the relative scattering intensity between scattering and absorption. Three peaks appear in the optical spectrum, for which two main peaks with a larger ratio of scattering efficiency are located at λ=402 nm and λ=660 nm. This implies that the underlying mechanism results in the peak of scattering efficiency differing from the plasmon resonance wavelength of absorption and scattering peaks. In addition, the scattering-efficiency peaks are wider than the absorption and scattering peaks. Experimentally, it is crucial to improve the scattering efficiency for cell and biomedical imaging applications with minimized absorption by adjusting its geometric parameters. Figure 2b shows the spectra of the silver–dielectric–gold three-layered nanoshell. Two main absorption peaks appear at about λ=414 nm and λ=730 nm, which can be attributed to the plasmon hybridization between a solid silver sphere and a gold nanoshell, indicating that the LSPR could be modulated by adjusting the material composition. Compared with the silver-dielectric-silver nanoshell, the scattering efficiency peaks of the silver–dielectric–gold nanoshell appeared at λ=454 nm, and λ=814 nm demonstrated that the scattering efficiency strongly depends on its material component. To help understand the effect of geometric parameters on the optical properties, Figure 2c,d show the spectra of gold-dielectric-gold and gold–dielectric–silver nanoshells. The spectra of the gold-dielectric-gold and gold–dielectric–silver nanoshells are similar to that of Figure 2a,b, where the scattering efficiency of the gold–dielectric–silver nanoshell is higher than that of the gold-dielectric-gold nanoshell. As a result, the scattering efficiency can be improved by adjusting the geometric parameters.

### 3.1. The Effect of Inner Core Radius on Scattering Efficiency

Figure 3 shows the effect of inner radius $r_{1}$ on the scattering efficiency of silver–dielectric–gold, where the inner silver radius $r_{1}$ is increased from $r_{1}=4$ nm to $r_{1}=16$ nm with $\left[r_{2},r_{3}\right]$ = [20 nm, 25 nm], and $\left[\varepsilon_{2},\varepsilon_{4}\right]=\left[2.0,4.0\right]$ for Figure 3a, $\left[\varepsilon_{2},\varepsilon_{4}\right]=\left[2.0,2.0\right]$ for Figure 3b, and $\left[\varepsilon_{2},\varepsilon_{4}\right]=\left[4.0,2.0\right]$ for Figure 3c respectively. The results show that the scattering efficiency of silver–dielectric–gold strongly depends on the inner core radius $r_{1}$, $\varepsilon_{2}$ and $\varepsilon_{4}$. As seen from Figure 3a, the intensity of each scattering-efficiency peak becomes intense and then reduces quickly with increasing the inner radius $r_{1}$. The intensity of the scattering-efficiency peak reduces first and reaches the minimum when the inner core radius $r_{1}=10$ nm; it then becomes intense quickly as the inner radius $r_{1}$ increases. The spectral width of the scattering efficiency is wider as the inner core radius increases, where the peak position takes place at a red-shift [11,14,18]. The inset of Figure 3a shows the effect of inner core radius $r_{1}$ on the scattering cross section, plotted as the function of wavelength; the scattering section of $|\omega_{-}^{+}\rangle$ mode takes place a red-shift, and its intensity reduces with increasing inner core radius $r_{1}$. Figure 3b,c show the influence of the middle dielectric function $\varepsilon_{2}$ and surrounding dielectric function $\varepsilon_{4}$ on scattering efficiency, where $\left[\varepsilon_{2},\varepsilon_{4}\right]=\left[2.0,2.0\right]$ in Figure 3b, and $\left[\varepsilon_{2},\varepsilon_{4}\right]=\left[4.0,2.0\right]$ in Figure 3c. Figure 3b,c show that the scattering efficiency strongly depends on the middle dielectric function $\varepsilon_{2}$ and surrounding dielectric function $\varepsilon_{4}$. The intensity becomes intense, and the spectral width becomes wider as the ratio of $\varepsilon_{2}/\varepsilon_{4}$ increases. In addition, the spectra of gold–dielectric–silver nanoshell are also shown in Figure 3d–f. Compared with Figure 3a–c, the scattering efficiencies of gold–dielectric–silver nanoshells are similar to those of silver–dielectric–gold nanoshells. The scattering efficiency strongly depends on the inner core radius $r_{1}$ and the ratio of $\varepsilon_{2}/\varepsilon_{4}$, which becomes intense, becomes wider, and has a red-shift with increasing $r_{1}$ and $\varepsilon_{2}/\varepsilon_{4}$. The scattering efficiency of gold–dielectric–silver nanoshell is higher than that of silver–dielectric–gold nanoshell under the same geometric parameter.

### 3.2. The Effect of Separated Dielectric Layer Thickness on Scattering Efficiency

To fully understand the influence of geometrical parameters on the scattering efficiency, we examine the effect of separated dielectric layer thickness $r_{2}-r_{1}$ on the scattering efficiency of silver–dielectric–gold nanoshells and gold–dielectric–silver nanoshells with different middle dielectric radius $r_{2}$ in Figure 4. The middle dielectric layer radius $r_{2}$ is increased from $r_{2}=7$ nm to $r_{2}=16$ nm, whereas the radii of the inner core and outer shell are fixed at $r_{1}=5$ nm and $r_{3}=25$ nm. Figure 4a shows that the intensity of scattering efficiency decreases quickly as the middle dielectric wall thickness increases from $r_{2}-r_{1}=2$ nm to $r_{2}-r_{1}=11$ nm with $\left[\varepsilon_{2},\varepsilon_{4}\right]=\left[4.0,2.0\right]$. The scattering-efficiency intensity remains at a constant when the middle dielectric thickness is large enough ($r_{2}-r_{1}=10$ nm). The scattering efficiency exhibits a blue shift with the increasing thickness of the middle dielectric layer. The inset of Figure 4a shows the scattering cross section has a blue-shift with increasing the thickness of $r_{2}-r_{1}$ due to plasmon hybridization [14,29]. The corresponding physical phenomenon implies that the thickness of the outer nanoshell reduces by increasing the middle dielectric layer radius. The influences of the middle dielectric function and surrounding medium function on scattering are also exhibited in Figure 4b,c with $\left[\varepsilon_{2},\varepsilon_{4}\right]=\left[2.0,2.0\right]$ and $\left[\varepsilon_{2},\varepsilon_{4}\right]=\left[4.0,2.0\right]$ respectively, the same as that in Figure 3a–c. In addition, the effects of middle dielectric thickness on the gold–dielectric–silver nanoshell are also given in Figure 4d–f, showing similar features to that in Figure 4a–c.

### 3.3. The Effect of Middle Dielectric Function on Scattering Efficiency

Since the dielectric function of the separated dielectric layer affects the electron oscillation, the scattering cross section and scattering efficiency depend on the dielectric function of the middle dielectric layer $\varepsilon_{2}$. Figure 5 shows the influences of the middle dielectric function $\varepsilon_{2}$ on the scattering efficiency with geometric parameter $\left[r_{1},r_{2},r_{3}\right]$ = [6 nm, 14 nm, 20 nm] and surrounding medium function $\varepsilon_{4}=4.0$, while varying the middle dielectric function $\varepsilon_{2}$ from $\varepsilon_{2}=1.0$ to $\varepsilon_{2}=6.0$. As seen from Figure 5a, the $|\omega_{-}^{-}\rangle$ mode has a redshift, and its intensity reduces quickly. $|\omega_{-}^{+}\rangle$ becomes larger quickly and then remains at a constant by increasing the middle dielectric layer $\varepsilon_{2}$. According to the plasmon hybridization theory [14,28], the cavity plasmon is affected by the cavity dielectric function and is reduced to lower energy by increasing the dielectric function $\varepsilon_{2}$. The coupling between the bonding shell plasmon mode and the inner core is reduced, in good agreement with the scattering cross-section in the inset of Figure 5a. Figure 5b examines the effect of $\varepsilon_{2}$ on the scattering efficiency of gold–dielectric–silver nanoshell. Both the intensity of $|\omega_{-}^{-}\rangle$ mode and $|\omega_{-}^{+}\rangle$ mode reduce quickly. The scattering-efficiency intensity of the gold–dielectric–silver nanoshell is much stronger than that of the silver–dielectric–gold nanoshell.

### 3.4. The Effect of Surrounding Dielectric Function on Scattering Efficiency

The surrounding medium dielectric function $\varepsilon_{4}$ also influences the scattering efficiency of bimetallic three-layered nanoshells; we explore the effect of surrounding medium dielectric function $\varepsilon_{4}$ on the scattering efficiency of bimetallic three-layered nanoshells, as shown in Figure 6, where the geometric parameter $\left[r_{1},r_{2},r_{3}\right]$ = [5 nm, 15 nm, 20 nm] and middle dielectric function $\varepsilon_{2}$ are fixed at $\varepsilon_{2}=2.0$. Figure 6a shows that the scattering efficiency of silver–dielectric–gold nanoshell increases by increasing the surrounding medium function $\varepsilon_{4}$ from $\varepsilon_{4}=1.0$ to $\varepsilon_{4}=6.0$, where $|\omega_{-}^{+}\rangle$ becomes intense quickly and $|\omega_{-}^{-}\rangle$ increases slowly, unlike that of Figure 5. The inset of scattering cross section is shown in Figure 6a, where $|\omega_{-}^{-}\rangle$ causes a distinct red-shift with increasing surrounding medium function $\varepsilon_{4}$ [15,19]. Due to the dielectric screening with a high refractive index, the induced charge on the outer gold nanoshell surface decreases, and then the induced charge on the inner surface of the silver core follows. Therefore, the electric repulsion force between the core and the outer nanoshell reduces. Thus, $|\omega_{-}^{-}\rangle$ mode exhibits a red shift due to the dielectric screening. Figure 6b also provides the scattering efficiency of the gold–dielectric–sliver nanoshell, compared with Figure 6a. Figure 6b shows that scattering efficiency is similar to that in Figure 6a, where both $|\omega_{-}^{+}\rangle$ and $|\omega_{-}^{-}\rangle$ become intense quickly. The scattering efficiency of the silver–dielectric–gold nanoshell is much stronger than that of the gold-dielectric-sliver nanoshell.

### 3.5. The Local Electric Field Intensity Distribution of Gold–Dielectric–Silver Nanoshells

Finally, in order to find the effect of geometrical parameters on scattering efficiency of bimetallic three-layered nanoshells, we calculate the local electric field intensity distributions of gold–dielectric–silver three-layered nanoshells with different geometrical parameters, middle dielectric layer function $\varepsilon_{2}$, and surrounding medium $\varepsilon_{4}$, where the direction of the incident electric field is parallel to the x-axis. Figure 7a,b show local electric field intensity distributions of gold–dielectric–silver nanoshell with LSPR wavelength λ = 444 nm and λ = 722 nm, respectively, where the geometrical parameters are $\left[r_{1},r_{2},r_{3}\right]$ = [10 nm, 15 nm, 25 nm] and $\left[\varepsilon_{2},\varepsilon_{4}\right]=\left[4.0,2.0\right]$. As shown in Figure 7a, we can see that the local electric field is concentrated in the inner gold core, while the local electric field intensity of silver nanoshell is very weaker at wavelength λ = 444 nm denotes as $|\omega_{-}^{+}\rangle$ mode. For $|\omega_{-}^{+}\rangle$ mode, the same kind of charges signed on the inner gold core and out of the silver nanoshell, which repulses in $|\omega_{-}^{+}\rangle$ mode. Figure 7b shows the local electric field focused in the middle dielectric layer, whereas the inner gold core and outer silver nanoshell are much weaker at wavelength λ=722 nm. There are different kinds of charges signed on the inner gold core and inner surface of the outer silver nanoshell, which attracts in $|\omega_{-}^{+}\rangle$ mode [15,19]. The scattering efficiency can be well improved with the larger inner core radius $r_{1}$, smaller middle dielectric layer function $\varepsilon_{2}$, and larger surrounding medium $\varepsilon_{4}$. Figure 7c,d provide local electric field intensity distributions of gold–dielectric–silver nanoshell with $\left[r_{1},r_{2},r_{3}\right]$ = [15 nm, 18 nm, 30 nm] and $\left[\varepsilon_{2},\varepsilon_{4}\right]=\left[1.0,5.0\right]$. Figure 7c,d show that the local electric field intensity is much stronger than that of Figure 7a,b; the decreasing middle dielectric layer leads to the surface induced charges increasing, and then the attraction leads to the local field intensity greatly increasing.

## 4. Conclusions

In conclusion, we have examined how the geometrical parameters, including the inner radius, the middle dielectric layer thickness, the middle dielectric function, and the surrounding medium dielectric function, affect the scattering efficiency of three-layered nanoshells by using the quasi-static theory. Our results have demonstrated that the scattering efficiency strongly depends on those geometric parameters. The intensity of the scattering efficiency can be improved by reducing the middle layer dielectric function $\varepsilon_{2}$ and increasing the surrounding medium function $\varepsilon_{4}$. As a result, a more robust scattering efficiency can be achieved by controlling the geometrical parameters and surrounding dielectric function. Since the scattering cross section of the homogeneous spherical core-shell particles is calculated analytically, it opens a way to understand the underlying physics by comparing the results with that by using the Mie scattering theory [30,34].

## Figures and Tables

**Figure 1 nanomaterials-12-03816-f001:**
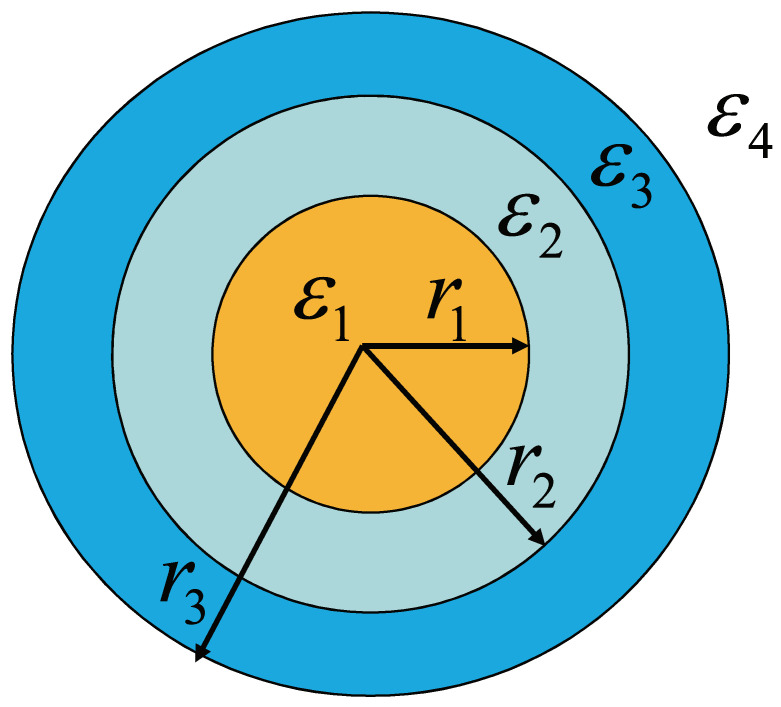
The geometry of bimetallic three-layered nanoshell. $r_{1},r_{2}$, and $r_{3}$ are the radii of inner core, middle dielectric layer, and outer nanoshell. $\varepsilon_{1}$, $\varepsilon_{2}$, $\varepsilon_{3}$, and $\varepsilon_{4}$ are the dielectric functions for the inner core, dielectric layer, outer nanoshell, and surrounding medium, respectively.

**Figure 2 nanomaterials-12-03816-f002:**
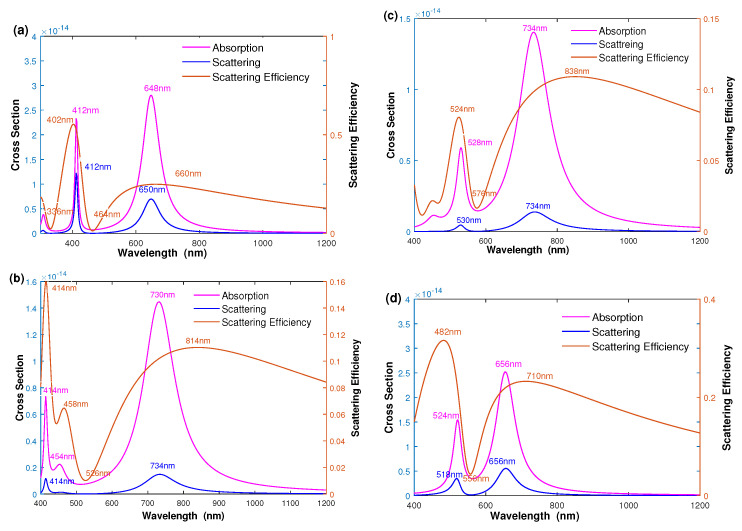
The spectra of absorption, scattering, and scattering efficiency are plotted as a function of wavelength. (**a**) Silver–dielectric–silver nanoshell, (**b**) silver–dielectric–gold nanoshell, (**c**) gold-dielectric-gold nanoshell, and (**d**) gold–dielectric–silver nanoshell.

**Figure 3 nanomaterials-12-03816-f003:**
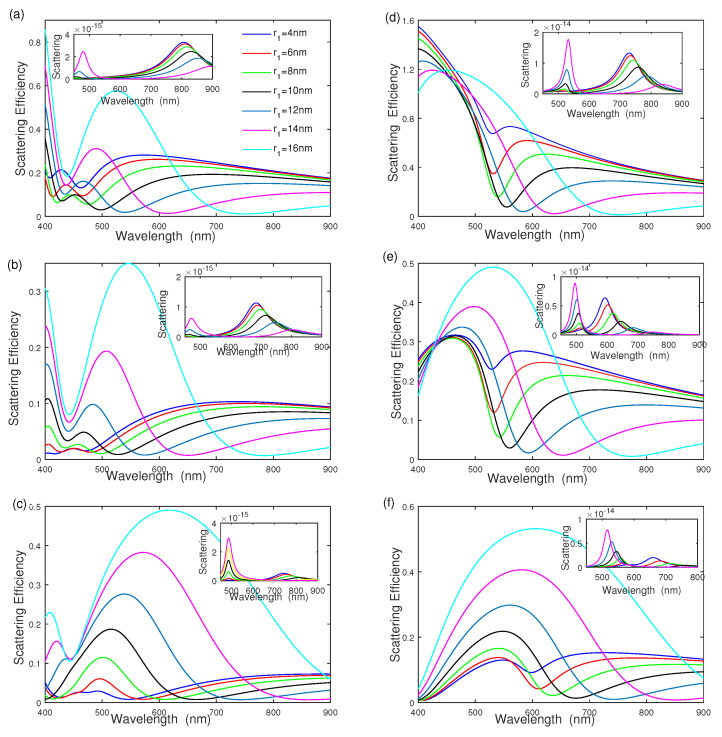
The effect of inner core radius $r_{1}$ on scattering efficiency and scattering. (**a**–**c**) Silver–dielectric–gold nanoshell with $\left[\varepsilon_{2},\varepsilon_{4}\right]=\left[4.0,2.0\right]$, $\left[\varepsilon_{2},\varepsilon_{4}\right]=\left[2.0,2.0\right]$, and $\left[\varepsilon_{2},\varepsilon_{4}\right]=\left[2.0,4.0\right]$; (**d**–**f**) gold–dielectric–silver nanoshell with $\left[\varepsilon_{2},\varepsilon_{4}\right]=\left[4.0,2.0\right]$, $\left[\varepsilon_{2},\varepsilon_{4}\right]=\left[2.0,2.0\right]$, and $\left[\varepsilon_{2},\varepsilon_{4}\right]=\left[2.0,4.0\right]$, respectively.

**Figure 4 nanomaterials-12-03816-f004:**
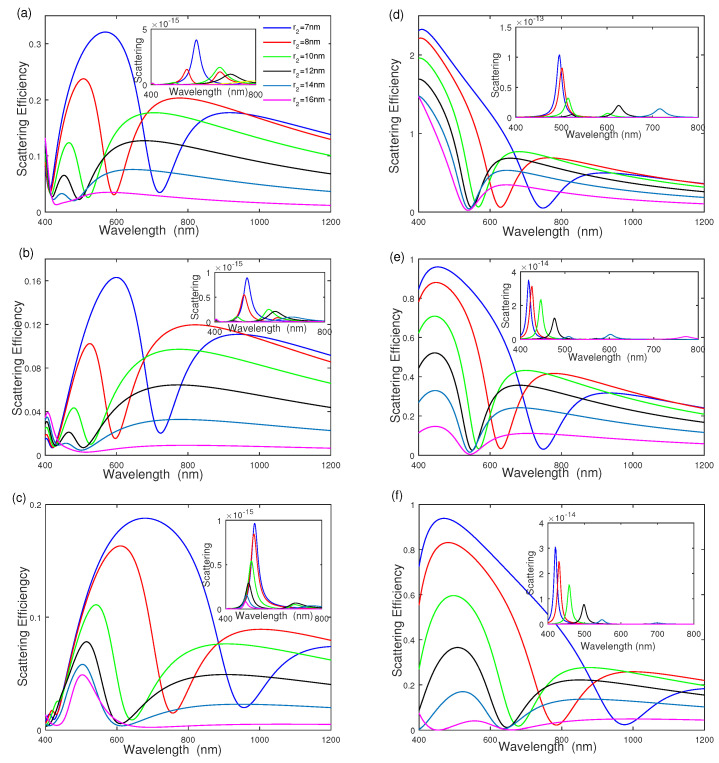
The effect of middle dielectric layer thickness $r_{2}-r_{1}$ on scattering efficiency and scattering. (**a**–**c**) Silver–dielectric–gold nanoshell with $\left[\varepsilon_{2},\varepsilon_{4}\right]=\left[4.0,2.0\right]$, $\left[\varepsilon_{2},\varepsilon_{4}\right]=\left[2.0,2.0\right]$, and $\left[\varepsilon_{2},\varepsilon_{4}\right]=\left[2.0,4.0\right]$. (**d**–**f**) Gold–dielectric–silver nanoshell with $\left[\varepsilon_{2},\varepsilon_{4}\right]=\left[4.0,2.0\right]$, $\left[\varepsilon_{2},\varepsilon_{4}\right]=\left[2.0,2.0\right]$, and $\left[\varepsilon_{2},\varepsilon_{4}\right]=\left[2.0,4.0\right]$, respectively.

**Figure 5 nanomaterials-12-03816-f005:**
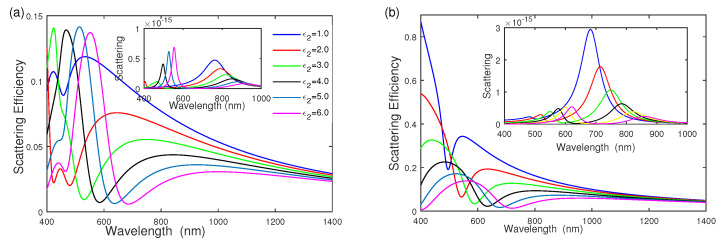
The effect of middle dielectric function $\varepsilon_{2}$ on scattering efficiency and scattering. (**a**) Silver–dielectric–gold nanoshell with $\varepsilon_{4}=4.0$, (**b**) gold–dielectric–silver nanoshell with $\varepsilon_{4}=4.0$, respectively.

**Figure 6 nanomaterials-12-03816-f006:**
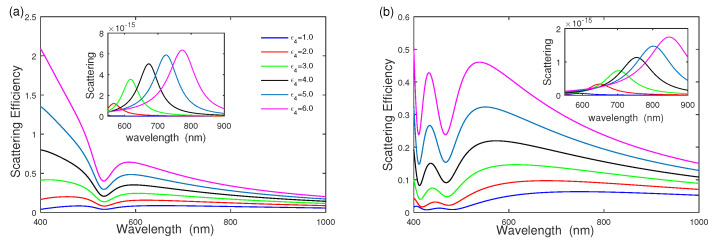
The effect of surrounding medium dielectric function $\varepsilon_{4}$ on scattering efficiency and scattering. (**a**) Silver–dielectric–gold nanoshell with $\varepsilon_{2}=2.0$, (**b**) gold–dielectric–silver nanoshell with $\varepsilon_{2}=2.0$, respectively.

**Figure 7 nanomaterials-12-03816-f007:**
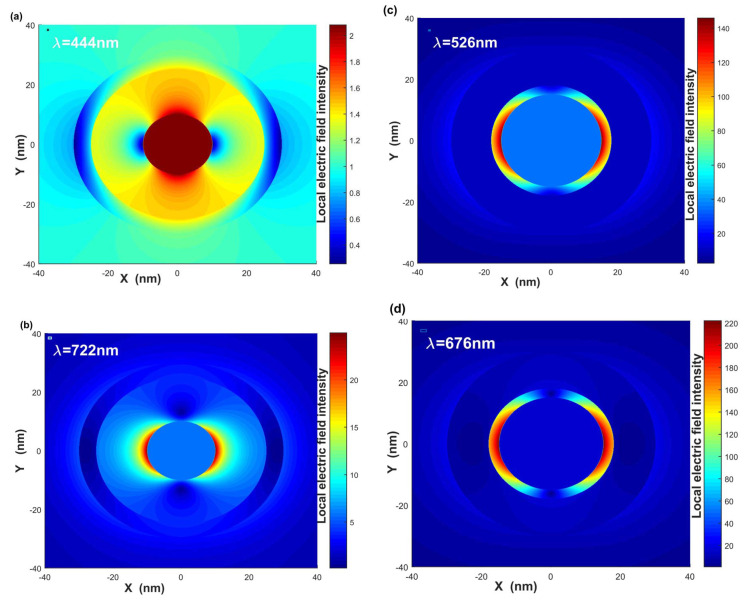
Local electric field intensity distributions of gold–dielectric–silver nanoshell with each LSPR wavelength. (**a**,**b**) $\left[r_{1},r_{2},r_{3}\right]$ = [10 nm, 15 nm, 20 nm], $\left[\varepsilon_{2},\varepsilon_{4}\right]=\left[4.0,2.0\right]$, and the local surface plasmon resonance wavelengths are at λ=444 nm and λ = 722 nm, respectively. (**c**,**d**) $[r_{1},r_{2},r_{3}]=$ [15 nm, 18 nm, 30 nm], $\left[\varepsilon_{2},\varepsilon_{4}\right]=\left[1.0,5.0\right]$, and the local surface plasmon resonance wavelengths are at λ=526 nm and λ = 676 nm.

## Data Availability

Not applicable.

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
