# Peer review of "Geometrical Parameter Effect on Plasmonic Scattering of Bimetallic Three-Layered Nanoshells"

_nanomaterials, 2022, doi:10.3390/nano12213816_

Round 1

Reviewer 1 Report

Comments to the authors,

   The authors have numerically calculated the ratio of scattering to absorption in bimetallic three layered nanoshells with different geometrical parameters such as inner core radius, shell thickness and dielectric functions using quasi-static theory. The calculated results provide useful information to develop biological sensors and photo-detectors with high performance. However the content of this article is not sufficient to be published in this journal at the present form. They should revise the manuscript as indicated below.

1.      The polarizability a in Eqs.(1, 2) is an essential formula in this article. So the authors should show the formula of a as reported in ref. 17 in appendix or supplement.

2.      They have not shown the damping constant g values that play important roles to calculate the scattering and absorption of light. They should show the values near line 62 in the sentence.

3.      We need the values of r1, r2, r3, e2 and e4 to calculate the results in Fig.2. They should show the values.

4.      The explanation of Fig.3 in the sentence (line 98~101, 111~115) is different from the figure caption of Fig.3. There is also no caption of Figs. 3 (e, f). What are the values of r2, r3 to calculate the results in Fig.3? They should revise them.

5.      r3-r2 in line 125 should be corrected to r2-r1. The explanation of Fig.4 in the sentence (line 138~141) is different from the figure caption of Fig.4. There is also no caption of Figs. 4 (e, f). They should revise them.

6.      What are the values of r1, r2, r3 to calculate the results in Fig.5? They should show the values.

7.      What are the values of r1, r2, r3 to calculate the results in Fig.6? They should show the values.

8.      Is [r1, r2]=[10, 15] on line 183 and in figure caption of Fig.7 correct? [r1, r2]=[10, 25] is considered correct. What is the value of r3 to calculate the results in Fig.7?

9.      The cross sectional shapes of nanoshells in Fig.7 look like ellipses, which should be corrected to circles. What is the direction of the incident electric field to calculate the results in Fig.7? They should revise them.

Author Response

Please find our response in attached. 

Author Response

Please find our response in attached

Reviewer 3 Report

The authors are encouraged to expand their references. There are many potential papers that could be cited, but these are just a few and the authors have many other papers that could also be considered. 

S. D. Campbell, R. W. Ziolkowski, "Impact of strong localization of the incident power density on the nano-amplifier characteristics of active coated nano-particles," Optics Communications, Vol. 285, No. 16, 2012.

S. D. Campbell and R. W. Ziolkowski, "Simultaneous Excitation of Electric and Magnetic Dipole Modes in a Resonant Core-Shell Particle at Infrared Frequencies to Achieve Minimal Backscattering," in IEEE Journal of Selected Topics in Quantum Electronics, vol. 19, no. 3, pp. 4700209-4700209, May-June 2013, Art no. 4700209, doi: 10.1109/JSTQE.2012.2227248.

J. A. Gordon and R. W. Ziolkowski, "The design and simulated performance of a coated nano-particle laser," Opt. Express 15, 2622-2653 (2007)

S. Arslanagic, R. W. Ziolkowski, and O. Breinbjerg, "Analytical and numerical investigation of the radiation from concentric metamaterial spheres excited by an electric Hertzian dipole", Radio Sci., 42, 2007. doi:10.1029/2007RS003663.

Author Response

Please find our response in attached.

Round 2

Reviewer 1 Report

The authors have revised their manuscript accoding to our comments.

This article in now worthy to be published in this journal at present form.

Reviewer 2 Report

The manuscript can be published as it is.

Reviewer 3 Report

The authors have addressed my previous comments.